# SeqSHAP: Subsequence Level Shapley Value Explanations for Sequential Predictions

## Abstract

With the increasing demands of interpretability in real-world applications, various methods for explainable artificial intelligence (XAI) have been proposed. However, most of them overlook the interpretability in sequential scenarios, which have a wide range of applications, e.g., online transactions and sequential recommendations. In this paper, we propose a Shapley value based explainer named SeqSHAP to explain the model predictions in sequential scenarios. Compared to existing methods, SeqSHAP provides more intuitive explanations at a subsequence level, which explicitly models the effect of contextual information among the related elements in a sequence. We propose to calculate subsequence-level feature attributions instead of element-wise attributions to utilize the information embedded in sequence structure, and provide a distribution-based segmentation method to obtain reasonable subsequences. Extensive experiments on two online transaction datasets from a real-world e-commerce platform show that the proposed method could provide valid and reliable explanations for sequential predictions.

## 1 Introduction

Sequential prediction tasks have a wide range of applications in real-world, e.g., **Online Transaction** (Wang et al., 2017; Zhang et al., 2018; Weber et al., 2018; Tam et al., 2019; Zhu et al., 2020; Chen & Lai, 2021) and **Sequential Recommendation** (Quadrana et al., 2017; Tang & Wang, 2018; Sun et al., 2019; Shen et al., 2021; Cui et al., 2022), since sequences contain continuous signals which are important for model predictions. With the development of deep learning technique, sequence-based models have achieved a desirable performance in recent years (Hidasi et al., 2015; Quadrana et al., 2017; Wang et al., 2017; Tang & Wang, 2018; Zhang et al., 2018; Sun et al., 2019; Zhu et al., 2020; Qiao & Wang, 2022). However, the complicated sequential data and increased model complexity make it hard for humans to understand the prediction of models. Indeed, for security and trust considerations, it is essential to develop effective explainable artificial intelligence (XAI) methods for sequence-based models in scenarios like fraud detection and medical care, so that end-users could understand how model predictions are produced with these complicated sequential data and models.

In recent years, considerable efforts have been made on the model explanation algorithms (Ribeiro et al., 2016; Shrikumar et al., 2017; Lundberg & Lee, 2017; Selvaraju et al., 2017; Wachter et al., 2017; Alvarez-Melis & Jaakkola, 2018; Mothilal et al., 2020; Slack et al., 2021; Ghalebikesabi et al., 2021; Ali et al., 2022). Among these works, feature attribution methods (Ribeiro et al., 2016; Shrikumar et al., 2016; 2017; Lundberg & Lee, 2017) are a popular family of post-hoc XAI methods. They calculate an attribution score for each feature to capture those important features for model predictions. However, most existing methods mainly pay attention to explain tabular data or images. And when dealing with the data and models in sequential scenarios, the complex input sequences make the element-wise explanations produced by these methods less explainable. The high-dimensional features and abundant interactions bring difficulty to existing element-wise XAI methods to provide explanations. Separately assigning attribution scores to individual feature cells in the sequence is not informative enough for users to understand the predictions. In addition, the great amount of features in a sequence could bring an extensive execution cost for existing methods, since the time complexity of them are mostly related to the number of features to be explained.

In this paper, we propose SeqSHAP, a Shapley value based method to explain model predictions in sequential scenarios. SeqSHAP provides explanations at a unique subsequence level, which is more intuitive in sequential scenarios for humans compared to the element-wise explanations. Meanwhile, we propose a distribution-based segmentation method to split the sequence into reasonable subsequences which utilizes the distribution information of sequential features. With obtained subsequences, we group the feature elements under each subsequence as independent units. Then Shapley value estimations for feature units are calculated, to capture the important features that strongly influence the model prediction.

Extensive experiments on two large-scale online transaction datasets collected from real-world are carried out. We analyze the local explanations produced by SeqSHAP and prove that our method provides intuitive explanations with meaningful subsequences, compared to existing feature attribution methods in sequential scenarios. Our contribution could be summarized as follows:

- We propose an effective XAI method to explain sequential predictions at a subsequence level, which is a unique and intuitive view in sequential scenarios.
- We propose a distribution-based segmentation method characterizing the distribution information of sequential features to capture the context information and obtain reasonable subsequences.
- Extensive experiments on two real-world transaction datasets are provided to evaluate the validity of our segmentation method and subsequence-level explanations produced by SeqSHAP.

## 2   BACKGROUND

In this section, we firstly introduce the task of explaining model predictions with sequential inputs. Then we introduce the background of SHAP (Lundberg & Lee, 2017), a popular interpretable framework based on Shapley values in game theory.

### 2.1   EXPLAINING PREDICTIONS WITH SEQUENTIAL DATA

Machine learning (ML) models for sequential prediction tasks have been widely applied in real-world applications (Hidasi et al., 2015; Tang & Wang, 2018; Sun et al., 2019; Zhu et al., 2020), since the historical behaviour records in a sequence contain valuable information for the prediction task. However, while different models with desirable performance are proposed, predictions are getting particularly difficult to explain due to the increasing model complexity, which blocks the application of new techniques in some scenarios requiring a high degree of interpretability. As a result, the demand of XAI methods in sequential domains is growing rapidly, as existing methods mostly focus on tabular data and are not suitable for data with sequence structure.

**Task Description**   In this paper, our task is building an interpreter $g$ to explain model predictions in sequential scenarios. Specifically, given a classifier $f$ and a sequence $X$ which could be formed as:
$$X = \{e_1, e_2, ..., e_T\}, \text{ where } e_t = \{x_1^t, x_2^t, ..., x_M^t\},$$
where $T$ is the length of sequence and $M$ is the number of features, $e_t \in \mathbb{R}^M$ represents the $t$-th element of sequence which has $M$ feature fields to describe it. The interpreter $g$ is expected to generate an explanation for the model prediction $\hat{y} = f(x) \in [0, 1]$. For the family of additive feature attribution methods, an element-wise explanation $\phi \in \mathbb{R}^{T \times M}$ assigns an importance score $\phi_{i,j}(1 \le i \le T, 1 \le j \le M)$ to the corresponding feature cell $x_j^i$ in the sequence $X$, which represents the influence of features on the model prediction.

### 2.2   SHAPLEY VALUE BASED EXPLANATIONS

**SH**apley **A**dditive ex**P**lanation, termed as SHAP (Lundberg & Lee, 2017), is a popular framework to explain model predictions based on the Shapley value in game theory. Through summarizing previous methods (Lipovetsky & Conklin, 2001; Štrumbelj & Kononenko, 2014; Bach et al., 2015; Datta et al., 2016; Ribeiro et al., 2016; Shrikumar et al., 2017), SHAP builds an additive explanation model $g$ as:

$$g(z) = \phi_0 + \sum_{i=1}^{M} \phi_i z_i, \tag{1}$$

where $M$ is the number of features, $z \in R^M$ is simplified features in a binary feature space, and an attribution score $\phi_i$ is assigned to each participating feature by solving Shapley values in a designed cooperative game. As long as there has been a lot of research and methods on SHAP's properties and applications in recent years (Sundararajan & Najmi, 2020; Frye et al., 2020; Zhang et al., 2020; Slack et al., 2020; Kumar et al., 2021; Jethani et al., 2021; Covert & Lee, 2021; Bento et al., 2021; Watson, 2022), here we mainly introduce KernelSHAP (Lundberg & Lee, 2017) which is most relevant to our work.

**KernelSHAP**  KernelSHAP is a model-agnostic explainer for local predictions which adopts the same objective function as the classic feature attribution method **LIME** (Ribeiro et al., 2016) shown in Eq. (2), while adjusting the choice of several settings to satisfy three desirable properties. Given a classifier $f$ and an input sample $x$, the objective function could be solved using weighted linear regression with the loss function $L$ in Eq. (3), where $h_x$ is a mapping function that maps simplified features $z$ to the original input feature space, and $\pi_x$ is a weighting kernel. The solution $\phi = \{\phi_i \mid i \in \{1, 2, \ldots, M\}\}$ provides an estimation of Shapley values for the input features.

$$\xi = \arg\min_{g} L(f, g, \pi_z) + \Omega(g) \tag{2}$$

$$L(f, g, \pi_z) = \sum_{z \in Z} [f(h_x(z) - g(z)]^2 \pi_x(z) \tag{3}$$

KernelSHAP treats each feature independently and calculate the attribution score using the weighted sum of feature's marginal contributions. It works well for tabular data since there is less context information, and the calculation of marginal contribution could partly model the interactions among features. However, for the case of sequential data, various contextual information is embedded in the sequence. And when explaining sequences, KernelSHAP has to adopt a Monte Carlo sampling strategy which sacrifices the precision of Shapley values to reduce the computational complexity, since the number of features in a sequence is too large to build a power set and calculate marginal contributions for all features. Accordingly, abundant contextual information could be ignored, and the element-wise explanation by KernelSHAP is less reliable.

## 3  SEQSHAP

In this section we introduce our method SeqSHAP which could provide more intuitive explanations for sequential predictions. Firstly, we discuss the motivation and advantage of explaining sequential predictions at the subsequence level. Then we propose a segmentation method to obtain reasonable subsequences from the input sequence. Finally, the process of generating subsequence level explanations with SeqSHAP is given.

### 3.1  EXPLAINING SEQUENTIAL PREDICTIONS AT SUBSEQUENCE LEVEL

A sequence is a stack of events that happened in a range of time, interactions of features among neighboring events often contains some hidden patterns (e.g., continuously changing, alternating, and recurring fields), which can be captured by ML models and has a significant impact on the model predictions. We note that applying attribution methods to a set of individual cells of the sequence (i.e., the cell-level explanation), which returns an importance matrix with the same shape as the input sequence: $G_{cell} = \begin{pmatrix} g_{1,1} & \cdots & g_{1,M} \\ g_{2,1} & \cdots & g_{2,M} \\ \vdots & \ddots & \vdots \\ g_{T,1} & \cdots & g_{T,M} \end{pmatrix}$ , can not explicitly model the effects of this interaction. And in practical applications, it is also difficult for end-users to understand the prediction with such an importance matrix, since humans tend to make predictions based on finding abnormal patterns rather than single feature cells in sequences.

We are inspired by the concept of session in recommendation system, which means several operations by a user over a short period of time. Now that a session could represent the user characteristics

during this time period, we attempt to group related neighbouring events in the sequence as sessions, and adopt XAI methods on them to explain the sequential data. By splitting the sequence into several subsequences and calculating importance scores for each feature under each subsequence (i.e., the subsequence-level explanation), explanations are provided as $G_{subseq} = \begin{pmatrix} g_{1,1} & \cdots & g_{1,M} \\ g_{2,1} & \cdots & g_{2,M} \\ \vdots & \ddots & \vdots \\ g_{K,1} & \cdots & g_{K,M} \end{pmatrix}$ , where $K \ll T$ is the number of subsequences split from the input sequence, and $g_{k,i}$ is the importance score of the $i$-th feature field under $k$-th subsequence. In this way, feature interactions between adjacent events are taken as a unit to be explained using attribution methods. And the explanation with a smaller shape provides a clearer guide for end-users to focus on the important areas in the sequence.

## 3.2 A DISTRIBUTION-BASED SEGMENTATION METHOD

As mentioned above, we hope those patterns of features can be explained as grouped units, through splitting the sequence into several subsequences. So the problem becomes how to split sequence properly while ensuring the events that make up a pattern can be grouped into the same subsequence. Simply split the sequence randomly or split with a fixed window could easily separate the related events and break the patterns, and the subsequences obtained are meaningless.

Here we suppose those hidden patterns and contextual information can be viewed as a specific distribution of the features, and propose a distribution-based segmentation method to get reasonable subsequences from the sequence. We attempt to maximize the distribution discrepancy among adjacent subsequences, in order to make adjacent subsequences include different context information. Firstly, the events happened within a specific time range are grouped as units $s_i (1 \leq i \leq k)$ and these units make up the initial set $S_{init}$ waiting to be segmented:

$$
\begin{aligned}
S_{init} &= \{s_1, s_2, \ldots, s_k\} \\
&= \{\{e_1, \ldots, e_{n_1}\}, \{e_{n_1+1}, \ldots, e_{n_2}\}, \ldots, \{e_{n_{k-1}+1}, \ldots, e_{n_k}\}\} \\
&\forall\, i \in [1, k], ts(e_{n_i}) - ts(e_{n_{i-1}+1}) \leq w\ , \delta_1 \leq |s_i| \leq \delta_2,
\end{aligned}
\tag{4}
$$

where $k$ is the number of grouped units, $w$ is the size of time window, $ts(e)$ is the scaled timestamp feature of event $e$, $\delta_1$ and $\delta_2$ are defined to limit the size of subsequence. Then, we insert split points into $S_{init}$ gradually, the point that maximizes a metric function will be chosen as the split point of the current round. The segmentation process is shown in algorithm 1:

---

**Algorithm 1** Distribution-based segmentation

---

**Input:** Initial set $S_{init}$, subsequence amount $K$, Metric function $D$
1: $S \leftarrow \{S_{init}\} = \{\{s_1, s_2, \ldots, s_k\}\}$, Split points $P \leftarrow \phi$
2: **while** $|S| < K$ **do**
3:  $d_{max} \leftarrow 0, p \leftarrow 0, S_p \leftarrow \phi$
4:  **for** $i \leftarrow 1$ to $|S_{init}| - 1$ **do**
5:   **if** $i \notin P$ **then**
6:    $P' \leftarrow Sort(P + i)$          $\triangleright$ Add point $i$ to $P$ temporarily
7:    $S' \leftarrow \{\{s_1, \ldots, s_{P'[1]}\}, \{s_{P'[1]+1}, \ldots, s_{P'[2]}\}, \ldots, \{s_{P'[-1]+1}, \ldots, s_k\}\}$
8:    $d \leftarrow D(S')$            $\triangleright$ Calculate the metric
9:    **if** $d > d_{max}$ **then**
10:     $p \leftarrow i, d_{max} \leftarrow d, S_p \leftarrow S'$
11:    **end if**
12:   **end if**
13:  **end for**
14:  $P \leftarrow P + p, S \leftarrow S_p$      $\triangleright$ Update split points and segmented subsequences
15: **end while**
**Output:** Segmented subsequences $S$

---

The metric function $D$ we design is shown in Eq. (5), where the input $S_p$ is the set of subsequences obtained after the latest insert at index $p$, $f_{dist}$ is a distance function measuring the distribution discrepancy between two subsequences in $S_p$ (e.g., MMD (Gretton et al., 2012), KL-divergence), $|s_i^p|$ and $|s_j^p|$ are used to limit the size of subsequences. And $m$ is the size of the measuring window

which determines how many neighbouring subsequences should be included to calculate the distance with the current subsequence. Our purpose is to distinguish the subsequences under different distribution, to capture the related events into a unit. The process of segmentation stops when the number of subsequences segmented reaches a given parameter $K < |S_{init}|$.

$$D(S_p) = \sum_{i=1}^{|S_p|-1} \sum_{j=min(i-m,1)}^{min(i+m,|S_p|)} \frac{f_{dist}(s_i^p, s_j^p)}{\sqrt{|s_i^p| * |s_j^p|}}, \tag{5}$$

### 3.3 PROVIDING EXPLANATIONS WITH SEQSHAP

With a sequence divided into $K$ subsequences, feature matrix of sequence $X \in R^{T \times M}$ could be formed as $X' \in R^{K \times M}$. Each row in $X'$ corresponds to a subsequence contains several events and $N = [n_1, n_2, \ldots, n_K]$ is the number of events in subsequences. The background values $B = [\overline{x_1}, \overline{x_2}, \ldots, \overline{x_M}]$ are sampled with average feature values in the dataset, to fill the absent features as uninformative feature values for the computation of Shapley values.

When explaining a sequence $X'$ directly using KernelSHAP, the explanation model $g^k$ is shown in Eq. (6). It takes $K * M$ feature units in $X'$ to build the coalition game, and the large feature space could bring an obvious loss of the precision using Monte Carlo sampling strategy to approximate the Shapley values, as mentioned in Subsection 2.2. To reduce the computational cost and the loss of precision, our method SeqSHAP calculates the subsequence-level Shapley values with two stages. For the first stage, we build a feature-level explanation model $g^f$ like TimeSHAP (Bento et al., 2021) as shown in Eq. (7), where each feature field of the sequence is taken as a unit and the feature-level Shapley values $\phi^f \in R^M$ are calculated with KernelSHAP.

$$f(h_{X'}(z)) \approx g^k(z^k) = \phi_0^k + \sum_{i=1}^{K} \sum_{j=1}^{M} \phi_{i,j}^k z_{i,j}^k. \tag{6}$$

$$f(h_{X'}(z)) \approx g^f(z^f) = \phi_0^f + \sum_{j=1}^{M} \phi_j^f z_j^f. \tag{7}$$

The simplified features $z^f \in \{0,1\}^M$ in Eq. (7) could be treated as a coalition of $z^k$ in Eq. (6), i.e., $z_j^f = 0$ is equivalent to $\forall i \in [1, K], z_{i,j}^k = 0$, since each column of $X'$ is taken as a unit in $g^f$. And there is $\phi_0^k = f(h_{X'}(z_0^k)) \approx f(h_{X'}(z_0^f)) = \phi_0^f$, since $z_0^k$ is equivalent to $z_0^f$ while all the features are absent.Therefore, the feature-level explanation $\phi^f$ is actually an estimation of the sum of features' Shapley values in KernelSHAP:

$$\phi_j^f \approx \sum_{i=1}^{K} \phi_{i,j}^k, 1 \le j \le M, \tag{8}$$

The second stage of SeqSHAP is shown in Algorithm 2, with feature-level explanations $\phi^f$, we traverse $M$ feature fields of $X'$ to provide subsequence-level explanations. For the case of $j$-th field, the explanation model $g_j^{seq}$ is built with the candidate feature set $S_j = \{x'_{1,j}, x'_{2,j}, \ldots, x'_{K,j}\}$, where $x'_{i,j}$ represents the $i$-th subsequence of $j$-th feature field. The simplified feature $z_j^{seq} \in R^K$ corresponds to the presence of subsequences in $S_j$ and the mapping function $h_{x'}$ is defined to map $z_j^{seq}$ to the original feature space:

$$h_{x'}(z_j^{seq}) = \begin{bmatrix} x'_{1,1} & \cdots & h_{x'}(z_{1,j}^{seq}) & \cdots & x'_{1,M} \\ x'_{2,1} & \cdots & h_{x'}(z_{2,j}^{seq}) & \cdots & x'_{2,M} \\ \vdots & \vdots & \vdots & \vdots & \vdots \\ x'_{K,1} & \cdots & h_{x'}(z_{K,j}^{seq}) & \cdots & x'_{K,M} \end{bmatrix}, h_{x'}(z_{i,j}^{seq}) = \begin{cases} x'_{i,j} & if \ z_{i,j}^{seq} = 1 \\ [\overline{x_j}]^{n_i} & if \ z_{i,j}^{seq} = 0 \end{cases}. \tag{9}$$

Thus when it is $j$-th feature's turn, other features in the mapping result are original input features, and the simplified features $z_j^{seq}$ determines the subsequences under $j$-th feature whether to be retained with original input values or replaced by uninformative feature values. Notably, each element

$x'_{i,j}$ in the matrix represents a subsequence of $j$-th feature with the shape of $n_i$, and the replacement of element is achieved by filling the same number of uninformative values as $n_i$. Afterwards, KernelSHAP is applied to solve the explanation model:

$$g_j^{seq}(z_j^{seq}) = \phi_{0,j}^{seq} + \sum_{i=1}^{K} \phi_{i,j}^{seq} z_{i,j}^{seq}, \quad \phi_{0,j}^{seq} = \phi_0^f + \sum_{i \neq j} \phi_i^f, \tag{10}$$

the definition of $\phi_{0,j}^{seq}$ is based on Eq. (6) and Eq. (7), since other feature fields are taken as the background and will not change during sampling coalitions, the effect of them should be added into the bias part of the explanation model. Hence the calculated explanations satisfy the property of Eq. (11), which maintains the consistency of explanation among features. Finally, sub-sequence level explanation $\phi$ is obtained by repeating the process $M$ times, and each element $\phi_j^{seq} \in R^K$ is the Shapley value for $K$ subsequences under $j$-th feature.

$$\sum_{i}^{K} \phi_{i,j}^{seq} \approx \phi_j^f. \tag{11}$$

---

**Algorithm 2** Subsequence-level explanation

---

**Input:** sequence $X' \in R^{K \times M}$, classifier $f$, feature-level attributions $\phi^f$
 1: **for** $j \leftarrow 1$ to $M$ **do**
 2:      $S_j \leftarrow [x'_{1,j}, x'_{2,j}, \ldots, x'_{K,j}]$                    ▷ Candidate feature set
 3:      $z^j \leftarrow [z_1^j, z_2^j, \ldots, z_K^j] \subseteq \{0,1\}^K$          ▷ Simplified features
 4:      $h_x(z^j) \leftarrow$ Equation(9)                         ▷ mapping function
 5:      $g(z^j) \leftarrow$ Equation(10)                       ▷ Explanation model
 6:      $\phi^j = [\phi_1^j, \phi_2^j, \ldots, \phi_K^j] \leftarrow KernelSHAP(f, g, S_j, z^j, h_x)$
 7: **end for**
**Output:** $\phi = [\phi^1, \phi^2, \ldots, \phi^M]$

---

Through calculating Shapley values with two stages, SeqSHAP reduce the size of feature space significantly compared to applying KernelSHAP directly to the sequence $X'$. Indeed, SeqSHAP acquires fewer perturbed samples to calculate the Shapley values and could provide subsequence-level explanations for sequence data more precisely with lower computation cost.

## 4   EXPERIMENTS

To evaluate our method, experiments on two online transaction datasets from an e-commerce platform in real world are carried out. The task of XAI methods is to provide local explanations for sequence-based fraud detection models, which are used to help end-users understand the model predictions, as described in Subsection 2.1.

### 4.1   EXPERIMENTAL SETUP

Our datasets are collected from a large-scale e-commerce platform, consisting of approximately 1.1M (**Dataset A**) and 1.6M (**Dataset B**) samples separately. Each sample $X$ in these datasets is tabular and corresponds to a sequence of one user's historical operation records on the platform ordered by the time, as formed in 2.1. An operation record, which is called an event here, includes $M$ features to describe the details of the event. The details of the datasets are shown in Table 1.

For considerations of privacy, the feature names are encoded into several types according to the content they describe (e.g., location, time, account, description of transactions). For dataset A, a given classifier $f_A$ predicts whether the last event of the sequence $e_T$ is a fraudulent transaction, based on the historical events $e[1 : T - 1]$. And for dataset B the classifier $f_B$ predicts whether the user is a fraudster based on the operation records $e[1 : T]$ happened recently. In our experiments, we choose RNN-based models as the classifier for the prediction tasks, which is built with an embedding layer, two LSTM layers and several feed-forward layers. Both models $f_A$ and $f_B$ are fit optimally on two datasets separately. Additionally, the detailed settings of parameters in our segmentation method could be found in Appendix A.

Table 1: Dataset details

|  | Dataset A | Dataset B |
|---|---|---|
| Max Events $T$ | 150 | 150 |
| Feature Fields $M$ | 33 | 30 |
| Categorical Features | 18 | 20 |
| Numerical Features | 15 | 10 |
| Samples | 1.1M | 1.6M |
| Classifier | $f_A$ | $f_B$ |

## 4.2 COMPARATIVE EXPERIMENTS

**Metric of Feature Attributions.** In our quantitative experiments, we take feature removal experiments to evaluate the local explanations produced by different feature attribution methods. Since the ground truth of an explanation is hard to get, we remove the top $\alpha\%$ elements in the ordered explanations and observe the difference of model predictions. A large change of the predicted score means that the explanation does capture those important features for model predictions. We define the metric $Dr_\alpha$ as Eq. (12), where $x$ is the origin input and $x'$ is the perturbed input obtained by replacing the top $\alpha\%$ elements in the explanation $\phi$ with uninformative feature values.

$$Dr_\alpha = \frac{1}{N} \sum_{i=1}^{N} \frac{|f(x') - f(x)|}{f(x)}. \tag{12}$$

**Experiments on Segmentation Methods.** We compare the performance of different segmentation methods to show our distribution-based method could capture relative events and generate reasonable subsequences. The subsequence-level attributions are calculated as local explanations with the subsequences obtained from different segmentation methods. And the metric $Dr_\alpha$ are used to compare the perfomance of explanations. The result is shown in Table 2, where segmentation method *Uniform* means split the input sequence with a fixed window of size $\lfloor T/K \rfloor$, and *Random* means split the sequence into $K$ subsequences randomly. *Ours(KL)* and *Ours(MMD)* are our distribution-based method using KL-divergence and MMD distance as the distance function $f_{dist}$, separately. We choose the drop rate $\alpha$ from $\{1\%, 1.5\%, 2\%, 2.5\%\}$ and calculate the mean of changes of model predictions after removing top $\alpha$ elements in the explanations. The result shows that the subsequences split using our distribution-based segmentation method are more reasonable for the subsequence-level explanations, while the contextual information helpful for the model predictions are better grouped in subsequences and removing the top elements could bring significant changes to model predictions.

Table 2: $Dr_\alpha$ of different segmentation methods

| Method | Dataset A | | | | Dataset B | | | |
|---|---|---|---|---|---|---|---|---|
|  | $Dr_{1\%}$ | $Dr_{1.5\%}$ | $Dr_{2\%}$ | $Dr_{2.5\%}$ | $Dr_{1\%}$ | $Dr_{1.5\%}$ | $Dr_{2\%}$ | $Dr_{2.5\%}$ |
| *Random* | 0.1304 | 0.2128 | 0.2612 | 0.2921 | 0.2943 | 0.3677 | 0.4015 | 0.5823 |
| *Uniform* | 0.2215 | 0.2707 | 0.3384 | 0.5168 | 0.3471 | 0.4162 | 0.5699 | 0.6881 |
| *Ours(KL)* | 0.2870 | 0.3375 | 0.4135 | 0.5938 | 0.3667 | 0.4134 | 0.5589 | 0.6918 |
| *Ours(MMD)* | **0.3006** | **0.3828** | **0.4941** | **0.6487** | **0.4382** | **0.4826** | **0.6204** | **0.7284** |

**Experiments on Local Explanations.** To prove that the subsequence-level explanations from SeqSHAP outperforms the element-wise explanations from existing feature attribution methods, we choose two popular feature attribution methods, KernelSHAP (Lundberg & Lee, 2017) and LIME (Ribeiro et al., 2016) as our baselines. We compare the explanations produced by SeqSHAP and the baselines with the metric $Dr_\alpha$ mentioned above. Since both baseline methods provide element-wise explanations, we drop the top $\lceil T * M * \alpha\% \rceil$ elements in the explanations from baselines, and $\lceil K * M * \alpha\% \rceil$ elements in SeqSHAP when calculating $Dr_\alpha$ for fairness. The amount of perturbed samples for KernelSHAP and LIME are set to $64K$ to make a trade-off between the efficiency and precision. As shown in Table 3, the performance of KernelSHAP and LIME are similar since

they apply the same objective function (Eq. (2)) with different kernel weights for perturbed samples (Eq. (3)). Our method outperforms the baselines in most cases on two datasets, which means the important feature subsequences are accurately captured and assigned with higher attribution scores by SeqSHAP, and the removal of them makes a large difference to the model predictions.

Table 3: $Dr_\alpha$ of different attribution methods

| Method | Dataset A | | | | Dataset B | | | |
|---|---|---|---|---|---|---|---|---|
| | $Dr_{1\%}$ | $Dr_{1.5\%}$ | $Dr_{2\%}$ | $Dr_{2.5\%}$ | $Dr_{1\%}$ | $Dr_{1.5\%}$ | $Dr_{2\%}$ | $Dr_{2.5\%}$ |
| *LIME* | 0.2728 | 0.3249 | 0.4551 | 0.6028 | 0.4248 | **0.4865** | 0.5471 | 0.6029 |
| *KernelSHAP* | **0.3015** | 0.3595 | 0.4498 | 0.6261 | 0.4161 | 0.4709 | 0.5954 | 0.6422 |
| *Ours(MMD)* | 0.3006 | **0.3828** | **0.4941** | **0.6487** | **0.4382** | 0.4826 | **0.6204** | **0.7284** |

## 4.3 CASE STUDY

In this section, we analyze several local explanation cases with sequence samples to show the effect of subsequence-level explanations from SeqSHAP. Firstly, we choose a positive sample from **Dataset A** whose latest event $e_T$ in sequence is fraudulent, which is predicted correctly with a large confidence by the RNN-based classifier $f_A$. We generate local explanations for the sample applying KernelSHAP and SeqSHAP separately. As shown in Figure 1, we visualize the feature attributions $\phi^{kernel} \in R^{T \times M}$ and $\phi^{seq} \in R^{K \times M}$, where $T$ is the length of sequence and $K$ is the number of subsequences obtained with our segmentation method. For simplicity we drop the feature fields where all cells under the feature have a smaller absolute attribution score than a threshold $\delta_a$, and mask the cells whose absolute attribution score $\phi_{i,j}$ is smaller than another threshold $\delta_b$ (the grey cells in Figure 1). The darker the cell in heatmaps, the larger attribution score it gets which means more important for the model prediction. It is obviously that the element-wise explanation $\phi^{kernel}$ in Figure 1(a) is not intuitive for users to understand with so many events and feature fields, even if we have dropped those less important elements. And for the case of subsequence-level explanation $\phi^{seq}$ in Figure 1(b), each cell represents the importance of a subsequence including several related events under a feature field. It is more intuitive with less subsequences ($K \ll T$) for users to understand which parts of the sequence promotes the classifier to make such a prediction.

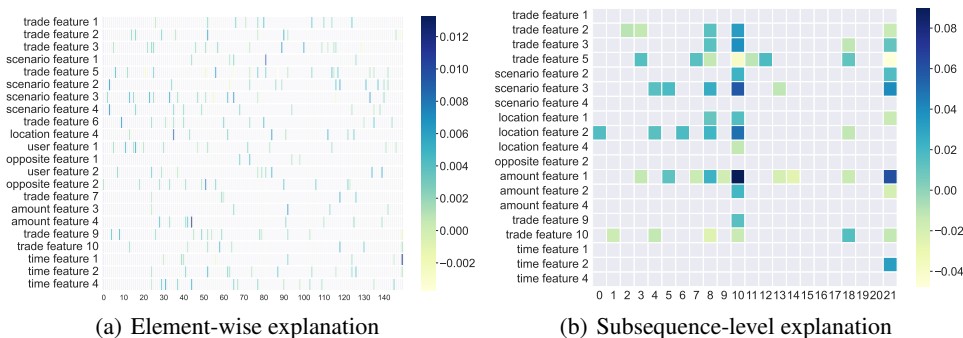

| (a) Element-wise explanation | (b) Subsequence-level explanation |
|---|---|

Figure 1: Local explanations with KernelSHAP and SeqSHAP

Our method SeqSHAP provides explanations at subsequence-level and the feature attribution results $\phi^{seq} \in R^{K \times M}$ assign an importance score to each subsequence for each feature field. Through analysing the distribution of attributions along different axes, higher level explanations could be obtained. Figure 2 visualize the higher level explanations in two different views, capturing the importance of subsequences and features separately. Figure 2(a) shows the importance of subsequences where the Shapley values of features under the $i$-th subsequence are plotted along the y-axis, which could help locate the abnormal subsequenes . Figure 2(b) provides a feature-level importance explanation, which could help identify the influential features in sequence.

As mentioned in Subsection 3.1, SeqSHAP explains a sequence at subsequence-level to capture those contextual information and hidden patterns included in the related neighbouring events of the

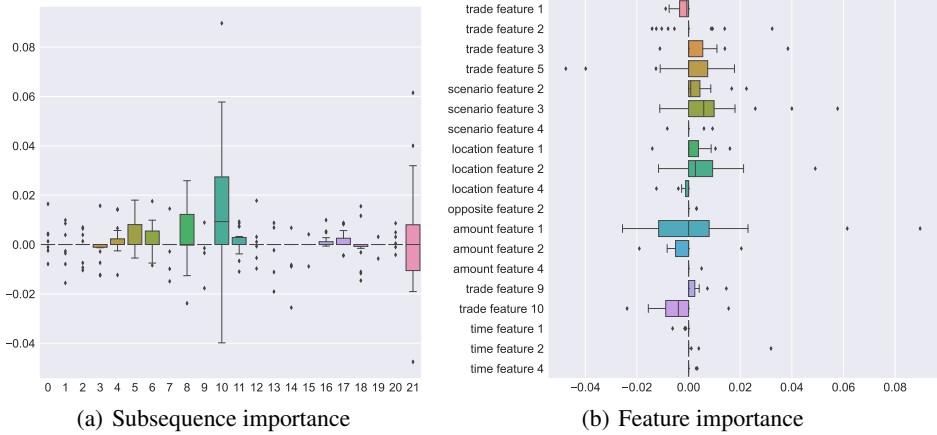

(a) Subsequence importance      (b) Feature importance

Figure 2: Two higher level explanations

sequence. We analyze the elements in explanation $\phi^{seq}$ with large attribution scores and summarize the corresponding input features of $\phi^{seq}_{i,j}$. As shown in Figure 3, the top elements with largest Shapley values are labeled with the summary of corresponding features in the sequence sample. Multiple abnormal patterns are discovered in the 10-th subsequence, including several transactions with large amount, variable locations and uncommon operations. Other subsequences with patterns like transactions happened at midnight and repetitive failed transactions are also assigned with high scores. Indeed, our segmentation method could provide reasonable subsequences including explicit feature patterns, and SeqSHAP is applied to find the important subsequences for model predictions.

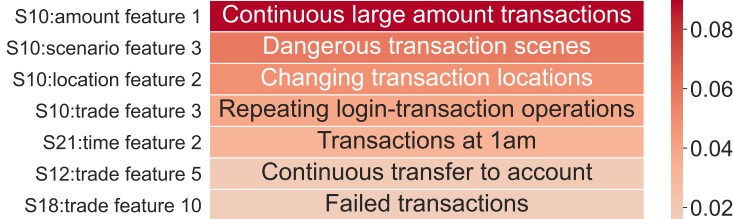

Figure 3: Semantic patterns included in the important subsequences with higher Shapley values.

## 5 CONCLUSION

For security and trust considerations in real world, the increasing need for explainable AI promotes the study on post-hoc feature attribution methods. However existing XAI methods mostly overlook the interpretability in sequential scenarios which has a wide range of applications. The widely used element-wise explanations which assign importance scores to all feature cells in a sequence is not intuitive for end-users to understand and could cause a huge reduction of the precision of explanations.

In this work, we propose SeqSHAP to explain sequential predictions at the subsequence-level, a unique view for feature attribution methods. We provide a distribution-based segmentation method to obtain reasonable subsequences to capture the hidden patterns and contextual information among neighbouring events. Through evaluating the explanations on online transaction datatsets collected from real-world, SeqSHAP is proven to be able to generate reliable Shapley value explanations for sequential data. With the user studies looking into the explanations and input features, the subsequence-level explanations are confirmed to be aligned with human concepts and could help users find out the abnormal patterns in the sequence that significantly influence the model predictions.

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

## A  PARAMETERS SETTING OF SEGMENTATION METHOD

For the first step we divide input sequence into units $S_{init}$ (Eq. (6)), the size of time window is defined as $w = max(\frac{ts(e_0)}{100}, 0.01)$, where $ts(e_0)$ is the scaled time interval between the earliest event and latest event. $\delta_1 = 3$ and $\delta_2 = 8$ are set to limit the size of initial units. Maximum mean discrepancy(MMD) Gretton et al. (2012) is chosen as the distance function $f_d ist$ to split the sequence(Eq. (5)), the embedding representation of events obtained from the given model $f$ are used to calculate the distribution distance. The number of sessions $K$ is defined as $min(max(10, \frac{|S_{init}|}{2}), 30)$, for the precision and efficiency of computing Shapley values.

