# OpenReview forum: "SeqSHAP: Subsequence Level Shapley Value Explanations for Sequential Predictions"
_ICLR.cc/2023/Conference — Submitted to ICLR 2023_

### Official Review · Reviewer_EXsb · 2022-10-23

**Confidence:** 4
**Correctness:** 2
**Technical Novelty And Significance:** 2
**Empirical Novelty And Significance:** 2
**Recommendation:** 5

**Clarity, Quality, Novelty And Reproducibility:**

Some problems are not clarified well. The related work is not discussed well. See above Strength And Weaknesses for details.

**Strength And Weaknesses:**

Strong points:
1. The motivation of subsequence-level explanation is clear.

Weak points:
1. The proposed distribution-based segmentation is given in Algorithm 1 without any explanation, which is hard to understand. Some notions and technical details are unclear. See details.
2. Sequential prediction has abundant applications, such as online transaction, sequential recommendation. However, the authors only validate their method in a binary classification fraud detection task with simple LSTM networks. It does not apply the proposed method to explain other reprehensive tasks and models in sequential prediction, such as BERT4Rec.
3. Some problems are not clarified well. For example, the authors point out the high-dimensional features and abundant interactions in sequences. But, they do not give specific data or evidence to explain the difficulty. In addition, the number of features in experimental datasets also cannot reflect the challenge.
4. The related work is not discussed well. Many methods are proposed to explain DNN in sequence data, such as [A1] and [A2]. What is the difference between the proposed method and these methods.
5. The authors point out the weakness of element-wise explanations. However, many works also consider the effect of multiple input variables [A1, A3]. What is the difference between multiple variables explanation and subsequence-level explanation?

[A1] Zhang, Hao, et al. "Interpreting multivariate shapley interactions in dnns." Proceedings of the AAAI Conference on Artificial Intelligence. Vol. 35. No. 12. 2021.
[A2] Bento, João, et al. "TimeSHAP: Explaining recurrent models through sequence perturbations." Proceedings of the 27th ACM SIGKDD Conference on Knowledge Discovery & Data Mining. 2021.
[A3] Murdoch, W. J.; Liu, P. J.; and Yu, B. 2018. Beyond word importance: Contextual decomposition to extract interactions from LSTMs. arXiv:1801.05453 .


Details：
* In Eq. (4), what is the relationship between the time window $w$ and the limit of subsequence size $\delta_{1}$ and $\delta_{2}$？
* In Algorithm 1, what is the meaning of the function Sort()?
* How to select an appropriate subsequence amount K?

**Summary Of The Paper:**

This paper focuses on the interpretability of deep learning in sequential scenarios. To this end, the authors propose to explain sequential predictions at the subsequence-level, and a distribution-based segmentation method.

**Summary Of The Review:**

Some problems are not clarified well. The related work is not discussed well. See above Strength And Weaknesses for details.

---

### Official Review · Reviewer_Sgob · 2022-10-24

**Confidence:** 3
**Correctness:** 3
**Technical Novelty And Significance:** 3
**Empirical Novelty And Significance:** 3
**Recommendation:** 5

**Clarity, Quality, Novelty And Reproducibility:**

The paper is well written and clear, even if sometimes there are phrases too long and difficult to read. The methodology is well presented, and I particularly appreciated the availability of the pseudo-code for every part of the methodology. Just a small typo, in the presentation of the segmentation strategy, the authors use the variable K in 2 different moments: for the general segmentation and for the segmentation only based on time, hence the first step only.
Regarding the novelty of the work, it does seem to me a small novelty: the KernelSHAP method is used everywhere and the novelty resides only in the segmentation at the beginning. However, it clearly is an interesting application.

**Strength And Weaknesses:**

+ The idea presented is interesting: the sequences are often long and may also have several features to take into account, hence an explanation on them may be difficult to understand.
+ The authors provide the pseudo-code for the subsequence extraction, as well as for the application of the KernelSHAP, in a clear manner.
+ The idea of considering subsequences may overcome these problems, providing an explanation that is easier to understand for the end user.
- However, the authors stress the fact that "for the end user the importance matrix is too difficult to understand, since humans tend to make predictions based on finding abnormal patterns rather than single features in sequence’" but the citation is missing: did the authors make some experiments to justify this belief? Did the authors read it in some other article?
- Regarding the methodology part, there are a few typos and at some point the authors use the letter K to indicate different things.
+ The experiments are well-written and correlated by several plots, which allows for an easier understanding of the metrics and properties tested.
- However, the datasets used are only 2 and from the same kind of data, e.g. e-commerce.


**Summary Of The Paper:**

In this paper, the authors present SeqSHAP: a method to provide SHAP explanations for sequences exploiting subsequences. The methodology starts by segmenting the sequence in input. The authors proposed a method for segmenting the sequences based on the distributions of the sequence elements. Once they obtain the subsequences, they group the feature elements under each subsequence as independent. On this result, they apply KernelSHAP.
They provide the pseudo-code of the segmentation procedure, which is based on different steps, considering both the time and a metric function based on MMD or KL-divergence, and they validate their methodology by conducting experiments on 2 datasets from e-commerce platforms. They compare the goodness of the explanations obtained by applying insertion/deletion methodologies.


**Summary Of The Review:**

The paper provides an interesting idea, straightforward. There are a few typos, but overall, the paper is well written and the authors provide the pseudo code for each part of the methodology. The experiments are conducted	on 2 multivariate datasets from e-commerce, and show good performance w.r.t. the simple application of the KernelSHAP method, even if they reach results which are not so good overall (the best performance is achieved with the segmentation method at 2.5%, but the values are around 0.64 and if I recall correctly this metric highlight a good feature importance with values closer to 1). Also the idea of showing the semantic patterns is also interesting and may be a starting point to improve this methodology.
However, I also have a concern about the validity of the method proposed: in the methodology the authors clearly stated that they took inspiration from recommender systems in the implementation of the segmentation procedure, which is based on a split on the time and then they apply a metric function (MMD or KL divergence). Then the authors test the method on e-commerce datasets, which are close to recommender systems. Given this setting, it seems to me that this approach may be feasible for this kind of data, but may fail for other kinds of sequences, such as energy consumption or trajectories. Have the authors tried other kinds of data?

---

### Official Review · Reviewer_z5Gv · 2022-10-24

**Confidence:** 4
**Correctness:** 3
**Technical Novelty And Significance:** 3
**Empirical Novelty And Significance:** 2
**Recommendation:** 6

**Clarity, Quality, Novelty And Reproducibility:**

- Not very clear, and seem to jump to steps without clear justifications (not even appendix)
- Confusing notations where for example in eq(10) "i" is used for features but before was used for subsequences.
- Experiments are not transparent and opaque and don't even have toy experiments to illustrate.
- Experiments also do not have standard deviations which makes me question the paper results itself. (please provide 2std in the table. If you do not I assume they are overlapping and hence have no stats significance)
- Not reproducible at all as the dataset is not even disclosed.


**Strength And Weaknesses:**

I will start with the strengths of this paper:

- Interesting setting to consider Shapley values in the sequential data setting.
- An interesting way to find the subsequences using MMD
- New way to think about how to interpret time series data i.e. in terms of subsequences which is more useful to a practitioner than each time step separately.
- Show improved performance compared to naive alternatives.

Next, the weakness of the paper and what needs to be clarified
- The clarity of the paper is not very good, firstly, when trying to find the subsequences, how many time does one need to compute the MMD? Intuitively speaking the worst case, the choices of splitting points could be enormous i.e. searching through every configuration of K subsequences. How do you deal with this and could you please add the number of MMD evaluations required (on average?). I might have missed this, please direct me to the place where you mention this.
- The paper in section 3.3 states that the algorithm has two stages. if I understand correctly, stage one aims to look at the Shapley values irrespective of the subsequences and then the second stage takes into account the subsequences. (looking at eq 7 here). Is that correct?
- Next in eq10 the authors claim to "apply KernelSHAP". could the authors please elaborate on this and write down the actual optimization objective to solve here? I am confused about how to compute these Shapley values. The way I understand it atm is that in standard Shapley values we have a binary mask where we remove subsets of features and thus are able to determine the shapley values of each feature. Here, we are removing sets of features i.e. subsequences in batches, hence obtaining the Shapley values for the subsequence?. I would appreciate it if the authors would write out the exact objective and also justify how their methods still follow the 3 axioms for Shapley in their setting. The main reason is, I see many \approx signs and I have no idea why they are approx and not equal.
- Experiments:
  -The authors only apply the model on proprietary datasets and have no synthetic toy experiments to show how their method works. Hence I am doubtful about how well the model actually performs. I highly suggest the authors run some synthetic data experiments to show how the methods work in a controlled environment where we know that feature 1-2 are important and 3-6 are not important at time step 1->k and vice versa for k+1-> T. That way the authors would also be able to verify how well the subsequence works. As it stands the authors seem to have no idea how well or what the subsequence even produces. All they know is that it is better than random and uniform. Hence I would highly recommend running some synthetic experiments to really illustrate how the method works through ablation.
- The next thing that has to be added is the sensitivity to K the authors should run a study varying over K to understand how sensitive the algorithm is to K. I assume at some point random/uniform will be just as good as your MMD/KL version. Please add these simple experiments on the toy experiments. In your dataset A/B what does the MMD produce? i.e. what is the distribution of the subsequence lengths? I assume there is no pattern as else the uniform would perform well as well?
- There are no standard deviations on the table of results and hence to my eyes useless and unreliable especially because there is no code. How many times were the experiments run? and what are the 2std? are they overlapping?
- How were the lengthscales of the MMD chosen? I see so many heuristics in the appendix that are simply not justified at all


These are my initial thoughts but I am more than happy to change my scores if the above have been addressed.


**Summary Of The Paper:**

This paper proposes a new way to compute Shapley values when presented with time series data. In particular for fraud detection in time series settings, understanding the reason behind the fraud detection algorithms is essential in many real-world settings. In this paper, the authors first proposed to split the time series into subsequences and then explain the prediction of the sequential models in terms of the features as well as the subsequence itself. In particular they propose a computationally efficient way to compute shapley values on each of each of the subsequences which thus allows a practitioner to understand which segment of the time series was most important. In addition, due to their setup they are also able to look into the importance of the features themselves as well.

They validate the idea on two unknown datasets and show superior performance on both datasets compared to naive approaches.


**Summary Of The Review:**

I have clearly outlined my concerns about the paper above and will be looking forward to hearing from the authors.
I am also more than happy to increase my score if the above have been addressed.

---

> ### Comment · Reviewer_z5Gv · 2022-12-11
> **Response**
>
> First of all sorry for the late reply as I was away for a personal matter and then sick for quite a while right after ...
>
> I have read through the rebuttal and appreciate the author's additional experiments, which is very much appreciated!
>
> The method seems to show some desirable properties in the sequential setting and has addressed most of my concerns.
>
> Hence I will raise my score for this paper and await also comments from other reviewers

---

### Official Review · Reviewer_nQqx · 2022-10-27

**Confidence:** 3
**Correctness:** 3
**Technical Novelty And Significance:** 2
**Empirical Novelty And Significance:** 3
**Recommendation:** 5

**Clarity, Quality, Novelty And Reproducibility:**

This paper is well presented with an interesting idea based on Shapley value. Since it’s mainly based on Shapley value technique, this may limit the originality contribution.

**Strength And Weaknesses:**

Strength:
In this work, the idea that splits a sequence into performing sub-sequences and subsequence-level explanation are reasonable. The proposed distribution-based segmentation method is also interesting and easy to understand. The user studies show the effectiveness of SeqSHAP.

Weakness:
This method is built upon Shapley value with certain modifications specific to sequential data, which compromises the technical novelty of this work. And the selection of parameters is not clear to me, which should influence the final performance. For example, how to choose an optimal K suitable for sequences with different lengths. Besides, the comparison methods in Table 2 are vanilla so that they not quite convincing. Maybe more stronger methods need to be considered. In addition, the baseline methods, i.e., LIME and KernelSHAP, used in Table 3 are proposed in 2016 and 2017 respectively. More recent methods should be considered as baselines, for example, “On locality of local explanation models, NeurIPS 2021”. Please address the weakness parts in the rebuttal.


**Summary Of The Paper:**

This work proposes SeqSHAP to study the interpretability of sequential data based on Shapley value. Instead of making element-wise explanations, SeqSHAP segments a sequence into sub-sequences, then uses Shapley value to characterize important features for a model decision. User studies were conducted to show the explainability ability of this method.

**Summary Of The Review:**

This work prompts explanations for sequential data, which can be complmentary to existing XAI methods. A sub-sequence level explanation method is proposed and validated in this work. However, the technical novelty aspect is somewhat limited to me, and require to be addressed during the author feedback period.

---

### Decision · Program_Chairs · 2023-01-20

**Decision:**

Reject

**Justification For Why Not Higher Score:**

* This method is built upon Shapley value with certain modifications specific to sequential data, which limits technical novelty.
* Reviewers note that the selection of parameters is not clear.
* Baseline methods are dated, so results are not quite convincing. Stronger and recently published methods should be considered, though that was partially addressed in the rebuttal.
* Reviewers also noted that sequential prediction has abundant applications. However, the new method is validated only in a binary classification fraud detection task with a simple LSTM network.

**Justification For Why Not Lower Score:**

N/A

**Metareview: Summary, Strengths And Weaknesses:**

This work presents SeqSHAP, a method to provide SHAP explanations for sequences exploiting subsequences based on the Shapley values. Instead of making element-wise explanations, SeqSHAP segments a sequence into sub-sequences based on the distributions of the sequence elements, then uses the Shapley value to characterize important features for a model decision. User studies were conducted to show the explainability ability of this method.

**Strengths:**
* Reviewers appreciated the idea of splitting a sequence into sub-sequences and subsequence-level explanations. The proposed distribution-based segmentation method is also interesting and easy to understand. The user studies show the effectiveness of SeqSHAP.
* The idea presented is interesting: the sequences are often long and may also have several features to take into account, hence an explanation of them may be difficult to understand.

**Weaknesses:**
* This method is built upon Shapley value with certain modifications specific to sequential data, which compromises the technical novelty of this work.
* Reviewers note that the selection of parameters is not clear. For example, how to choose an optimal K suitable for sequences with different lengths.
* Baseline methods in Table 2 are dated, so they are not quite convincing. Stronger and recently published methods should be considered, though that was partially addressed in the rebuttal.
* Reviewers also noted that sequential prediction has abundant applications, however, the new method is validated only in a binary classification fraud detection task with a simple LSTM network.